# Biological Activities and Chemical Profile of *Gentiana asclepiadea* and *Inula helenium* Ethanolic Extracts

**DOI:** 10.3390/molecules27113560

**Published:** 2022-05-31

**Authors:** Victoria Buza, Mihaela Niculae, Daniela Hanganu, Emoke Pall, Ramona Flavia Burtescu, Neli-Kinga Olah, Maria-Cătălina Matei-Lațiu, Ion Vlasiuc, Ilinca Iozon, Andrei Radu Szakacs, Irina Ielciu, Laura Cristina Ștefănuț

**Affiliations:** 1Department of Animal Physiology, Faculty of Veterinary Medicine, University of Agricultural Sciences and Veterinary Medicine Cluj-Napoca, 400372 Cluj-Napoca, Romania; catalina.matei@usamvcluj.ro (M.-C.M.-L.); ilincaiozon@yahoo.com (I.I.); cristina.stefanut@usamvcluj.ro (L.C.Ș.); 2Department of Infectious Diseases, Faculty of Veterinary Medicine, University of Agricultural Sciences and Veterinary Medicine Cluj-Napoca, 400372 Cluj-Napoca, Romania; pallemoke@gmail.com; 3Department of Pharmacognosy, Faculty of Pharmacy, “Iuliu Haţieganu” University of Medicine and Pharmacy, 400010 Cluj-Napoca, Romania; dhanganu@umfcluj.ro; 4SC PlantExtrakt SRL, 407059 Rădaia, Romania; ramona.burtescu@plantextrakt.ro (R.F.B.); neli.olah@plantextrakt.ro (N.-K.O.); 5Faculty of Pharmacy, Vasile Goldiș Western University of Arad, 310045 Arad, Romania; 6Department of Anatomy, Faculty of Veterinary Medicine, University of Agricultural Sciences and Veterinary Medicine Cluj-Napoca, 400374 Cluj-Napoca, Romania; ion.vlasiuc@usamvcluj.ro; 7Department of Animal Nutrition, Faculty of Veterinary Medicine, University of Agricultural Sciences and Veterinary Medicine Cluj-Napoca, 400374 Cluj-Napoca, Romania; andrei.szakacs@usamvcluj.ro; 8Department of Pharmaceutical Botany, Faculty of Pharmacy, “Iuliu Haţieganu” University of Medicine and Pharmacy, 400010 Cluj-Napoca, Romania; irina.ielciu@umfcluj.ro

**Keywords:** *Inula helenium* L., *Gentiana asclepiadea* L., ethanolic extracts, roots, chemical profile, antioxidant, antimicrobial, cytotoxic

## Abstract

This study aimed to investigate the antioxidant, antimicrobial, and cytotoxic potential of ethanolic extracts obtained from *Gentiana asclepiadea* L. and *Inula helenium* L. roots, in relation to their chemical composition. The total polyphenols, flavonoids, and phenolic acids were determined by spectrophotometric methods, while LC-MS analysis was used to evaluate the individual constituents. The antioxidant properties were tested using the FRAP and DPPH methods. The standard well diffusion and broth microdilution assays were carried out to establish in vitro antimicrobial efficacy and minimum inhibitory and bactericidal concentrations. The cytotoxicity was tested on rat intestinal epithelial cells using the MTT assay. The results pointed out important constituents such as secoiridoid glycoside (amarogentin), phenolic acids (caffeic acid, chlorogenic acid, *trans*-*p*-coumaric acid, salicylic acid), and flavonoids (apigenin, chrysin, luteolin, luteolin-7-*O*-glucoside, quercetin, rutoside, and naringenin) and promising antioxidant properties. The in vitro antimicrobial effect was noticed towards several pathogens (*Bacillus cereus* > *Staphylococcus aureus* > *Enterococcus faecalis* > *Salmonella typhimurium* and *Salmonella enteritidis* > *Escherichia coli*), with a pronounced bactericidal activity. Rat intestinal epithelial cell viability was not affected by the selected concentrations of these two extracts. These data support the ethnomedicinal recommendations of these species and highlight them as valuable sources of bioactive compounds.

## 1. Introduction

For ages, both humans and animals have been instinctively using plants for prevention and treatment of various diseases. In recent decades, the interest in herbal medicines (HMs) and traditional medicine has increased tremendously and led to the start of the “Return to Nature” trend [1]. Medicinal plants exhibit a wide variety of therapeutic effects, including anti-inflammatory, antioxidant, anti-proliferative, antiviral, and antimicrobial [2,3,4]. Currently, 25% of modern medicines are plant-derived, medicinal herbs and their bioactive compounds are regarded as safer and healthier substitutes to the synthetic drugs, especially in long-term use [1]. However, it was estimated that only 15% of species have been studied for their chemical composition, and around 6% for their therapeutic effects [5].

Gentianeae is the most species-rich tribe of the family Gentianaceae, comprising 974 species, from which around 360 belong to the *Gentiana* L. genus [6]. *Gentiana asclepiadea* L. (Willow gentian) is a perennial species belonging to the *Gentiana* genus, found mostly in regions with temperate climate and high altitudes in central, southern, and eastern Europe, Turkey, and Iran [6,7]. Based on its occurrence in nature and risk of extinction, its status varies depending on the region. Therefore, according to International Union for Conservation of Nature and Natural Resources (IUCN), *G*. *asclepiadea* is listed as strictly protected in Poland, vulnerable in Hungary and Germany, nearly threatened in Croatia, and of least concern in the rest of the countries [6,7,8]. In traditional medicine, the roots and rhizomes of *G*. *asclepiadea* are used for the treatment of digestive system disorders and hepatitis infections [3,8]. Similarly, in Romanian traditional, medicine, *G*. *asclepiadea* root tea is used as appetite stimulant, choleretic, anthelmintic and for the treatment of diarrhea [9]. Today, because of its biological effects and chemical composition, this species is commonly used as a substitute for *G. lutea* (yellow gentian), a species with high therapeutic value that is endangered and under protection in most European countries [7,8]. Scientific studies on the therapeutic effects of *G. asclepiadea* show that its roots possess antigenotoxic [3], antioxidant [2,3], hepatoprotective [10], antibiofilm [2], antibacterial [2,11], and prebiotic activities [11]. The secondary metabolites found in the chemical composition of *G. asclepiadea* roots and responsible for its biological activities are bitter secoiridoids glycosides (swertiamarin, gentiopicrin, amarogentin [12,13], sweroside [3]), flavonoids and xanthones (gentioside and gentisin) [13].

Asteraceae is one of the largest plant families that includes 1400–1700 genera and 24,000–35,000 species, representing 10% of all known flowering plant species [14]. The *Inula* L. genus comprises 78 to 100 species found in Europe, Asia, and Africa, known for the large therapeutic potential of their phytochemical compounds. *Inula helenium* L. (elecampane) is a widely spread herbaceous perennial species that belongs to the genus Inula [15,16]. Its collection from the spontaneous flora for medicinal purposes has led to a decrease in the populations found in eastern Europe, as indicated in the most recent report provided by IUCN [15]. In traditional medicine, the roots of *I. helenium* are used for the treatment of respiratory diseases (bronchitis, tuberculosis), gastrointestinal symptoms such as vomiting, diarrhea, abdominal pain, or poor appetite, associated with infectious or parasitic diseases and circulatory diseases [16,17]. *I. helenium* roots are also used externally in the treatment of wounds, pruritus, and rheumatic pain [16]. Currently, oral administration of *I. helenium* root tea is recommended in herbal medicine for the alleviation of respiratory symptoms, as a digestive tonic, choleretic, and vermifuge agent; as a topical application, this species is indicated in bacterial and fungal dermatitis and for skin disorders characterized by dry and itchy skin [16,17]. Recent studies have reported that roots of *I. helenium* exhibit in vitro anti-inflammatory, antioxidant, anti-proliferative, antimicrobial, antibacterial, anticandidal, prebiotic, and anthelmintic effects [4,18,19,20,21]. These proprieties could be attributed to its main secondary metabolites, such as sesquiterpene lactones (alantolactone and isoalantolactone), phenolic acids (gallic acid, caffeic acid, cinnamic acid, coumaric acid), and flavonoids (quercetin, myricetin, kaempferol, catechin) [16,22].

*G. asclepiadea* and *I. helenium* species are commonly used in traditional medicine for the treatment of digestive disorders. Their roots are believed to relieve abdominal pain, stimulate the gastrointestinal system and bile secretion, and exhibit anthelmintic activity [10,16]. Additionally, due to the presence of inulin in the roots of *I. helenium* and gentio-oligosaccharides (gentiobiose and gentianose) in *G. asclepiadea* roots, both can potentially influence the composition of gut microflora [11,21]. Based on the above-mentioned ethnomedicinal uses, oral administration of herbal medicines containing *I. helenium* and *G. asclepiadea* is recommended, their phytochemical constituents being absorbed at intestinal level.

Therefore, taking all the above-mentioned aspects into consideration, the study of *G. asclepiadea* and *I. helenium* species and their biological activities appears to be an important subject in order to support their traditional uses. In this context, the present study aimed to perform a comprehensive evaluation of the chemical composition of *Inula helenium* and *Gentiana asclepiadea* roots ethanolic extracts by spectrophotometry and LC-MS analysis, and of their in vitro antioxidant and antimicrobial efficacy. Moreover, the in vitro cytotoxicity was investigated using rat intestinal epithelial cell cultures, which to the best of our knowledge is the first report of the cytotoxic effect of *G. asclepiadea* and *I. helenium* ethanolic root extracts on primary intestinal cell culture, this being the main element of novelty and originality of the present study. Results obtained hereby may represent important aspects that bring further arguments to sustain the ethnomedicinal uses of these species in digestive disorders.

## 2. Results and Discussions

### 2.1. Quantification of Total Polyphenolic (TPC), Flavonoid (TFC), and Phenolic Acids (TPA) Content

Results obtained using spectrophotometrical methods for the quantification of TPC, TFC, and TPA content are presented in Table 1. Values obtained for these assays were significantly higher (*p* < 0.05) in the *I. helenium* ethanolic extract compared to the one obtained from *G. asclepiadea* (Table 1).

Results for the quantification of TPC in *I. helenium* roots (3.066 g GAE/100 g dry plant) were within the wide range of 1.5–71.24 mg GAE/g reported in other studies [22,23,24]. Based on existing research, the lowest value of TPC (1.5 mg GAE/g dry weight) was obtained by reflux extraction with 95% ethanol [23], and the highest value was obtained by Soxhlet extraction (71.24 mg GAE/g) [24]. Similarly, the TPC of *G. asclepiadea* roots (2.144 g GAE/100 g dry plant) was in agreement with the previously published data (5.64–146.64 mg GAE/g) [2,3,13].

However, in both plants, the TFC values were lower compared to other studies, the reported range for *I. helenium* roots being 9.32–50.0 mg RE/g [22,23] and for *G. asclepiadea* of 3.61–17.54 mg RE/g [2,3]. As per TPA content, the value for the ethanolic extract of *I. helenium* roots was significantly higher when compared to the *G. asclepiadea* root extract. This was further confirmed by differences in the concentrations of caffeic and chlorogenic acids, identified and quantified by LC-MS analysis in both tested extracts.

Large variations in these compound contents in extracts could be explained by the differences appearing in exogenous and endogenous factors, such as geographical, climatic conditions, exposure to UV-B radiation, harvesting period, plant age, genetic diversity, solvent, and extraction techniques [24].

### 2.2. Liquid Chromatography-Mass Spectrometry (LC-MS) Analysis

The identification and quantification of the chemical constituents of *G. asclepiadea* and *I. helenium* ethanolic extracts were achieved by a LC-MS method. The LC-MS method was validated for linearity, repeatability, limits of detection (LOD), and limits of quantification (LOQ). Major compounds identified in *G. asclepiadea* roots were amarogentin, apigenin, luteolin-7-*O*-glucoside, naringenin, and rutoside. In addition to the above-mentioned compounds, *G. asclepiadea* root extract contained *trans*-*p*-coumaric acid, caffeic acid, chlorogenic acid, luteolin, and unquantifiable amounts of salicylic acid, chrysin, and quercetin.

The LC/MS analysis of the *I. helenium* root ethanolic extract revealed that caffeic acid, chlorogenic acid, chrysin, luteolin, and hesperetin were the major compounds. *I. helenium* extract was also found to contain luteolin-*7*-*O*-glucoside and naringenin (Table 2).

The *G. asclepiadea* chromatogram of the major identified compounds is shown in Figure 1.

In total, 12 compounds were identified in the ethanolic extract of *G. asclepiadea*, of which one bitter secoiridoid glycoside (amarogentin), four phenolic acids (caffeic acid, chlorogenic acid, *trans*-*p*-coumaric acid, salicylic acid), and seven flavonoids (apigenin, chrysin, luteolin, luteolin-7-*O*-glucoside, quercetin, rutoside, and naringenin). In the case of *I. helenium* ethanolic extract, seven compounds were detected, namely two phenolic acids (caffeic acid, chlorogenic acid) and five flavonoid compounds (chrysin, luteolin, luteolin-7-*O*-glucoside, naringenin, and hesperetin).

Among the identified compounds, amarogentin was detected in *G. asclepiadea* roots extract in a concentration of 27.8 ± 0.3 µg/g. The presence of amarogentin in *G. asclepiadea* roots was previously reported by Szucs et al. [12], but only in trace amounts. This bitter secoiridoid glycoside was commonly isolated from various species of the genus Gentiana and Swertia, family Gentianaceae, such as *G. lutea*, *G. gelida*, *G. dinarica*, *S. chirayita*, *S. alternifolia*, *S. bimaculata*, *S. alata*, *S. nervosa*, or *S. ciliata* [13,25,26,27,28]. Recent studies pointed out a variety of biological effects for amarogenin, including antileishmanial, antioxidant, anti-diabetic, anticancerous, and antithrombotic activity [25,26,27,28].

Phenolic acids were identified in important amounts in the chemical profile of *G. asclepiadea* roots ethanolic extract. The maximum amount was found for the *trans*-*p*-coumaric acid (192.8 ± 1.0 µg/g dry vegetal material), followed by caffeic acid (169 ± 1.2 µg/g dry vegetal material), chlorogenic acid (33.4 ± 0.5 µg/g dry vegetal material), and salicylic acid (<LOQ). Two of these phenolic acids, namely chlorogenic acid and caffeic acid were also detected in high concentrations in *I. helenium* roots (2284.1 ± 11 and 234.0 ± 2.1 µg/g dry vegetal material, respectively). Both compounds have antioxidant and anti-inflammatory potential [29,30,31]. Furthermore, the chlorogenic acid was proven to modulate lipid metabolism [31].

From the flavonoids group, luteolin, luteolin-7-*O*-glucoside, chrysin, and naringenin were detected in both *G. asclepiadea* and *I. helenium* roots. The presence of luteolin-7-*O*-glucoside was previously reported in the chemical composition of *I. britannica*, *G. asclepiadea*, and *G. gelida* [13,32]. Similarly, luteolin was isolated from several species of genus Gentiana and Inula, including *G. arisanensis*, *G. veitchiorum*, *I. britannica*, and *I. viscosa* [32,33,34,35] and has been reported to possess antioxidant, anti-inflammatory, cardio-protective, neuroprotective, and antimicrobial effects [36]. The presence of naringenin was previously reported in *G. veitchiorum* flowers (0.12 mg/L), but in lower concentrations compared to *G. asclepiadea* roots [34]. According to recent studies, both naringenin and chrysin exhibit anticancer, anti-inflammatory, antioxidant, and antimicrobial activities [37,38].

Two flavonoids, namely rutoside and apigenin were among the major compounds identified only for the ethanolic extract derived from *G. asclepiadea* roots (30.8 ± 0.6 and 18.0 ± 0.7 µg/g dry vegetal material, respectively). Apigenin was earlier described as part of the chemical composition in aerial parts of *Gentiana* species, and roots of *G. asclepiadea*, *G. gelida*, and *G. paradoxa* [13,34]. Similar to other members of the flavonoid group, rutoside and apigenin have been reported to possess antioxidant, anti-inflammatory, and antimicrobial effects [34,39,40,41]. In addition, apigenin was associated with neurovascular protective effect, anti-diabetic activity, and the ability to suppress hepatic lipid accumulation [34,40].

### 2.3. Antioxidant Activity

The results of DPPH radical scavenging activity and ferric-reducing power (FRAP) of *G. asclepiadea* and *I. helenium* roots were expressed using the IC_50_ value (μg/mL) and μmol TE/100 mL extract, respectively (Table 3).

Although the presence of small to moderate quantities of reactive oxygen species (ROS) is regarded as indispensable for maintaining cellular homeostasis, their overproduction plays a key role in the pathogenesis of various inflammatory and neurodegenerative diseases [29]. In search of novel antioxidant agents, plants have been regarded as promising sources of bioactive compounds. Since bioactive compounds found in medicinal plants exert their antioxidant effects through multiple chemical mechanisms, with potential synergistic effects, two methods with different reaction mechanism were used to test the antioxidant activity of *I. helenium* and *G. asclepiadea* root extracts. DPPH radical scavenging (DPPH) assay is a method primarily based on single electron donating capacity of hydrophobic antioxidants, and hydrogen atom transfer as secondary mechanism [42]. The Ferric Reducing Antioxidant Power (FRAP) assay was used to detect the redox potential of hydrophilic compounds and is based on hydrogen atom transfer [42].

Based on the obtained results, both ethanolic extracts presented significant antioxidant capacities. This potential was significantly (*p* < 0.05) higher in case of *I. helenium*, which manifested greater ability to reduce ferric ions and showed a higher DPPH radical scavenging activity (629.04 μmol TE/100 mL and a IC_50_ value of 173.2). The *G. asclepiadea* extract exhibited a result of 145.23 μmol TE/100 mL extract for the FRAP assay and an IC_50_ value of 363.7 μg/mL for the DPPH assay, respectively. The DPPH assay results for *G. asclepiadea* and *I. helenium* root extracts fell within the large range previously reported in other studies, both plants exhibiting low to moderate DPPH radical scavenging activity [2,3,13,24,43]. Furthermore, the variations of FRAP and DPPH results were in accordance with the differences of plants total polyphenols, flavonoids, and phenolic acids content. This observation is supported also by the Pearson coefficients established for FRAP and TPC, TFC, and TPA content (*r*^2^ = 0.999, *r*^2^ = 0.996, and *r*^2^ = 0.999, respectively, *p* < 0.05). Statistical analysis pointed out significant negative correlation between DPPH values and the above-mentioned compounds (*r*^2^ = −0.999, *r*^2^ = −0.997, and *r*^2^ = −0.999, respectively, *p* < 0.05). Taking into account the DPPH results interpretation, radical scavenging activity also depends on the TPC, TFC, and TPA content. The direct correlation between the antioxidant capacity of plant extracts and their total phenolic and flavonoid concentrations, as well as other compounds with antioxidant activity, was confirmed in other similar studies [22,29,43,44].

Among the major compounds identified in *I. helenium* root extract by the LC-MS analysis, phenolic acid compounds chlorogenic acid and caffeic acid, and flavonoid compound luteolin were previously reported to exhibit concentration-dependent antioxidant activity by donating a hydrogen atom or electrons to free radicals [29,30,36]. Between the compounds detected in the ethanolic extract of *G. asclepiadea* roots, potent in vitro and in vivo antioxidant activity was previously reported for bitter secoiridoid glycoside (amarogentin), present in various species of genus Gentiana and Swertia [13,25,26,27,28,45]. Both in vitro and in vivo studies reported that amarogentin exhibits strong radical scavenging activity and possesses the ability to increase the radical-absorbing capacity of cells [28,45]. Another compound that might be responsible for *G. asclepiadea* root extract antioxidant activity is the flavonoid compound apigenin, that according to recent study exhibits remarkable ABTS and DPPH radical scavenging activity in vitro and regulates cholesterol metabolism in vivo [34]. Additionally, the antioxidant activity of *G. asclepiadea* roots can be attributed to the presence of phenolic acids (caffeic acid, chlorogenic acid), flavonoids (luteolin, naringenin, chrysin), and other unidentified compounds [2,3,36,37].

### 2.4. Antibacterial Activity

Results of the in vitro antimicrobial activity screening are displayed in Table 4 and Table 5.

Overall, both ethanolic extracts displayed in vitro antimicrobial activity (Table 4) against all selected bacterial strains. The potency of the antimicrobial efficacy varied depending mostly on the bacterial type, with a more intense inhibitory effect expressed against the Gram-positive species (*Bacillus cereus* > *Staphylococcus aureus* > *Enterococcus faecalis*) compared to the Gram-negative (*Salmonella enteritidis* = *Salmonella typhimurium* > *Escherichia coli*). Values obtained for the inhibition zone diameter ranged from 10.00 to 17.33 mm and 8.67 to 18.00 mm in the case of *G. asclepiadea* and *I. helenium* root extracts, respectively, thus significantly (*p* < 0.05) lower compared to those of gentamicin, the standard antibacterial agent. Two of the Gram-positive species, namely *Staphylococcus aureus* and *Bacillus cereus*, showed higher susceptibility when exposed to a combination of the two extracts, with diameters of the inhibition zone of 18.33 ± 0.47 and 21.00 ± 0.00, respectively. These values are similar to those induced by gentamicin (*p* > 0.05) (Table 4).

Values obtained for MIC, MBC, and the resulting MIC index obtained for the two extracts using the broth microdilution method are presented in Table 5. The bactericidal efficacy was clearly pointed out against all tested bacterial species (MBC/MIC ≤ 4).

The bacterial strains were selected given their antimicrobial resistance pattern and prevalence. Over recent decades, the research interest in antimicrobial resistance (AMR) has increased significantly. Recently, it was estimated that if no actions are taken, by 2050 AMR will be responsible for 10 million deaths each year [46]. Misleadingly referred by many as a “silent pandemic”, AMR has worsened since the outbreak of COVID-19 pandemic due to the widespread use of surface disinfectants, misuse, and overuse of antimicrobials [47]. Therefore, finding a safe therapeutic alternative to conventional drugs has become increasingly important for global health and the economy.

Previous studies documented the in vitro antimicrobial efficacy of *G. asclepiadea*- [2,48,49] and *I. helenium*-derived products [18,19,20,43,50], tested alone or in combination with different compounds.

An aqueous extract obtained from *I. helenium* was found active against *Bacillus mycoides* for MIC 5 mg/mL, and with synergistic activity combined with sodium nitrite and potassium sorbate against *Bacillus subtilis* and *Pseudomonas fluorescens* [19]. The in vitro anti-*Staphylococcus aureus* efficacy against both antibiotic-resistant and susceptible clinical isolates was documented for a hydroethanolic extract obtained from the rhizome and roots at concentrations between 0.9 and 9.0 mg/mL [50]. Similar in vitro anti-staphylococcal activity was reported for (hydro)ethanolic root extracts of *I. helenium* L. (elecampane) naturalized in Ireland supporting their traditional usage [20]. Additionally, these products demonstrated efficacy against other Gram-positive bacteria such as Group-A *Streptococcus pyogenes*, Group-B *Streptococcus agalactiae*, *Listeria monocytogenes*, and also Gram-negative *Escherichia faecalis* ATCC 29212 and *Escherichia coli*, as well as *Mycobacterium tuberculosis* H37Ra (ATCC 25177) [20]. In vitro antibacterial (*Staphylococcus aureus*, *Enterococcus faecalis*, *Escherichia coli*, *Pseudomonas aeruginosa*) and antifungal (*Candida albicans* and *Candida tropicalis*) properties were demonstrated by the agar dilution method in case of methanolic extracts obtained from three *Inula* species, *I. viscosa*, *I. helenium* ssp. *turcoracemosa*, and *I. montbretiana*, collected from different locations of Anatolia [43]. Moreover, a mixture of sesquiterpene lactones and essential oil extracted from *I. helenium* cultivated in Hungary exhibited considerable inhibitory effects against six species of fungi (*Candida albicans*, *Candida glabrata*, *Candida cruzei*, *Candida parapsilosis*, *Saccharomyces cerevisiae*, *Aspergilus niger*) and seven species of bacteria (*Staphylococcus aureus*, methicillin-resistant *Staphylococcus aureus*, *Streptococcus pyogenes*, *Bacillus subtilis*, *Escherichia coli*, *E. coli* D31, *Pseudomonas aeruginosa* [51]. For the Romanian cultivar, Diguță et al. [52] reported in vitro antimicrobial potential in the case of an ethanolic extract against veterinary strains of *Bacillus subtilis*, *Bacillus cereus*, *Enterococcus faecalis*, *Escherichia coli*, *Staphylococcus aureus*, *Candida albicans*, *C. parapsilosis*, *C. lipolytica*, and *Aspergillus niger*.

Certain *G. asclepiadea* extracts or their fractions prepared by maceration with methanol [49], water, ethanol, ethyl acetate, acetone, and diethyl ether demonstrated a better in vitro inhibitory efficacy against Gram-positive compared to Gram-negative bacteria [2,48]. At a concentration of 2.12 mg/mL, the aqueous extract of roots inhibited 50% of biofilm formation in case of *S. aureus* ATCC 25923 [2]. As for the active compounds responsible for the antimicrobial properties, the presence of xanthones [49] and of secoiridoid glycosides such as gentiopicroside, swertiamarin, and sweroside [12] appears to be relevant.

### 2.5. Cytotoxicity Assay

The results of MTT assay showed that *G. asclepiadea* roots extract (Figure 2a) did not exhibit any cytotoxic activity at the tested concentrations (0.0079–0.4726 μmol GAE/mL), cell viability ranging from 94.83 ± 3.58 to 74.89 ± 0.97%, respectively. The IC_50_ dose for *G. asclepiadea* extract was 1.1097 ± 0.028 μmol GAE/mL. At increasing concentrations (C6, C7, and C8), cell viability decreased statistically significantly compared to the untreated cells; however, this was not considered biologically significant. Additionally, a strong linear correlation was observed between the cell viability and the total polyphenolic content of *G. asclepiadea* extract (*r*^2^ = 0.7997).

Similarly, according to the study conducted by Hudecová et al. [53], the methanolic extract of willow gentian flowers did not exhibit any cytotoxic or genotoxic effect on monkey kidney cell line (COS 1) at concentrations ranging between 0.25 and 2.5 mg/mL. Additionally, previous studies showed that the methanolic extracts of *G. lutea* and *G. rigescens* protected the HepG2 (nontumorigenic human hepatoma) and THLE-2 (transformed human liver epithelial) cells from the cytotoxic effect of fatty acids and promoted the growth of HepG2 cells [54].

The results of MTT assay for the extract of *I. helenium* roots (Figure 2b) showed that at concentrations between 0.0135 and 0.5379 μmol GAE/mL, the ethanolic extract of *I. helenium* roots did not exhibit any cytotoxic activity, the cell viability ranging from 83.92 ± 6.37 to 70.11 ± 1.55%. However, at a concentration of 0.8068 μmol GAE/mL, a mild cytotoxic activity was observed, cell viability being 51.54 ± 4.68%. The IC_50_ dose of *I. helenium* extract was 0.9093 ± 0.016 μmol GAE/mL. Similar to the *G. asclepiadea* extract, a strong linear correlation was noticed between the cytotoxic effect of *I. helenium* extract and its TPC (*r*^2^ = 0.9485). At the highest tested concentration (C8), both extracts influenced the shape of the intestinal epithelial cells, causing them to become rounded and flat, without affecting their viability.

Similar to our results, alantolactone and isoalantolactone, the major bioactive compounds isolated from *I. helenium* roots, did not exert any cytotoxic effect on Caco-2 cells, widely used as intestinal epithelial cells model [55]. No cytotoxic effect of *I. helenium* extract was observed also in PBMC (peripheral blood mononuclear cells), RAW 264.7 macrophage cells, and BV-2 microglial cells [4,56,57]. However, on the human tumor cell lines (HT-29, MCF-7, Capan-2, G1), *I. helenium* root extract exhibited potent cytotoxic activity [4]. The differences in the cytotoxic activity of the extract on cancer and healthy cells can be explained by variances in their cellular metabolism.

## 3. Materials and Methods

### 3.1. Chemicals and Reagents

Methanol, formic acid, salicylic acid, and chrysin used for LC/MS analysis were purchased from Merck (Darmstadt, Germany). All other chemicals used as standards for LC-MS analysis were purchased form Phytolab (Vestenbergsgreuth, Germany). All microorganism strains were distributed by Oxoid Ltd. (Basingstok, Hampshire, UK), while the culture mediums, Mueller Hinton Broth and Mueller Hinton agar, were purchased from Merck (Darmstadt, Germany). Rat intestinal epithelial cells used for the cytotoxic potential were isolated from fetal donors. Collagenase type I, dispase type I, and Hanks’ balanced salt solutions used for the enzymatic digestion of intestine were purchased from Sigma-Aldrich (Darmstadt, Germany). DMEM medium used as isolation and propagation media was purchased from Sigma-Aldrich (Darmstadt, Germany). Fetal bovine serum (FCS) and antibiotic-antimycotic solution used to supplement the isolation and propagation media were purchased from Gibco Life Technologies (Paisley, UK). Non-essential amino acids and epidermal growth factor used to supplement propagation medium were purchased from Sigma-Aldrich (Darmstadt, Germany).

### 3.2. Plant Material and Extract Preparation

*Gentiana asclepiadea* and *Inula helenium* roots were purchased from an authorized herbal online store in Romania (AdServ SRL). The plant materials were identified by Lecturer Irina Ielciu, PhD, and voucher specimens species are deposited at the Department of Pharmaceutical Botany of the “Iuliu Haţieganu” University of Medicine and Pharmacy, Cluj-Napoca (Vouchers number 376–377). For the extract preparation, 5 g of dried roots were powdered at 450 µm particle size using a Gindomix GM 200 mechanical grinder (Retsch GmBH, Eragny, France) and mixed with 100 mL of 70% *v*/*v* ethanol. Moisture content was established at 12.5% for *G. asclepiadea* and at 10.5% for *I. helenium* using a Kern DLB Thermobalance (Kern&Sohn GmBH, Stuttgart, Germany). Resulting suspensions were vortexed thoroughly for 30 min and left in a dark place at room temperature to macerate for 10 days. The resulting suspensions were centrifuged at 4000 RPM for 10 min. The obtained 70% *v*/*v* ethanolic extract was filtered through grade 1 Whatman filter paper and stored in amber glass bottles at 4 °C.

### 3.3. Quantification of Total Phenolic, Flavonoid, and Phenolic Acids Content

The TPC was determined by a spectrophotometric method using Folin–Ciocâlteu reagent. Gallic acid was used as standard phenolic total, the result being expressed as g GAE per 100 g of dry plant material. Spectrophotometric determination of TFC was performed using aluminum chloride as chromogenic agent, and absorbance was measured at 430 nm. Moreover, rutoside was used as a standard reference solution for the construction of calibration curve, and the results were expressed as g RE per 100 g of dry material. TPA content was determined by spectrophotometric method, using Arnow reagent. The absorbance was determined at 500 nm, and TPA content was expressed as g CAE per 100 g of dry material. All the determinations were performed using a UV–V is spectrophotometer (Specord 200 Plus, Analytik Jena, Germany) [58,59,60,61].

### 3.4. Liquid Chromatography-Mass Spectrometry (LC-MS) Analysis

The LC/MS method was performed on a Shimadzu Nexera I LC/MS—8045 (Kyoto, Japan) UHPLC system equipped with a quaternary pump, autosampler, and an ESI probe and quadrupole rod mass spectrometer. The separation was carried out on a Luna C18 reversed phase column (150 mm × 4.6 mm × 3 μm, 100 Å), from Phenomenex (Torrance, CA, USA). The column was maintained at 40 °C during analysis.

The mobile phase (Table 6) was a gradient made from methanol and ultrapurified water prepared by Simplicity Ultra-Pure Water Purification System (Merck Millipore, Billerica, MA, USA). Formic acid was used as an eluent. The methanol and formic acid were of LC/MS grade. The used flow rate was of 0.5 mL/min. The total time of the analysis was 35 min.

The detection was performed on a quadrupole rod mass spectrometer operated with electrospray ionization (ESI), both in negative and positive MRM (multiple reaction monitoring) ion mode. The interface temperature was set at 300 °C. Gas nitrogen was used for vaporization and as drying at 30 psi, respectively, at 10 L/min. The capillary potential was set at +3000 V.

The references used for quantification can be found in Table 7, 1 μL of each reference at each concentration was injected. The identification was performed by comparison of the retention times, the MS spectra, and its transitions between the separated compounds and standards. The identification and quantification were performed based on the main transition from the MS spectra of the compound.

The LC-MS method was validated by evaluating linearity, precision, and accuracy according to International Conference on Harmonization guidelines (ICH). The LOD and LOQ were calculated after injecting a series of different concentrations for each standard. The extracts were assayed for precision under optimized conditions. The method accuracy was determined in duplicate by a recovery experiment. All samples and references were injected in triplicate.

For quantification purposes, the calibration curves were also determined (Appendix A). Calibration curves, equations, their correlation factors, and the determined limit of detection and quantification are presented in Table 8.

### 3.5. Antioxidant Activity

#### 3.5.1. Ferric-Reducing Antioxidant Power Assay (FRAP)

FRAP reagent was prepared by mixing 10 mM TPTZ solution in 40 mM HCl (2.5 mL), 20 mM FeCl_3_·6H_2_O solution (2.5 mL), and acetate buffer (25 mL, pH 3.6). For FRAP assay, 4 mL of plant extract was mixed with 1.8 mL of water, and 6 mL of FRAP reagent. In the negative control, the extract was replaced with 4 mL of water. The absorbance of obtained solutions was read at 450 nm, using Trolox as standard for the calibration curve (R2 = 0.992). The results of FRAP assay were expressed as μmol of Trolox Equivalents (TE) per 100 mL of extract [58,59,60,61]

#### 3.5.2. DPPH Radical Scavenging Activity Assay

DPPH assay was used to determine the antioxidant potential of *I. helenium* and *G. asclepiadea* root extracts. For the preparation of DPPH solution, 10 mg of DPPH was weighted and dissolved in 100 mL methanol. For each tested plant extract, a serial dilution was prepared by mixing 0.25, 0.5, 0.75, 1.0, 1.25, 1.5, 1.75, and 2 mL of extract with methanol, to obtain a final volume of 4 mL. Then, 2.0 mL of DPPH methanolic solution was added to prepare sample dilutions and the final reaction mixtures were incubated at 40 °C for 30 min. For the negative control, plant extract was replaced with 2 mL of DPPH methanolic solution. Absorbance was measured at 517 nm and the extract DPPH radical scavenging activity was calculated using the following formula:(1)$$(\%)\;\mathrm{Inhibition}=\frac{\mathrm{Absorbance}\;\mathrm{of}\;\mathrm{control}\;-\;\mathrm{Absorbance}\;\mathrm{of}\;\mathrm{sample}}{\mathrm{Absorbance}\;\mathrm{of}\;\mathrm{control}}\;\times 100\%$$

The results of the DPPH assay were expressed as IC_50_ value (μg/mL), representing the concentration of antioxidant capable of reducing the DPPH radical concentration by half [58,59,60,61].

### 3.6. Antibacterial Activity

The in vitro antimicrobial potential was screened by agar well-diffusion assay, a modified EUCAST (European Committee on Antimicrobial Susceptibility Testing) [62] disk-diffusion method. Six reference strains were included *Staphylococcus aureus* ATCC 25923, *Bacillus cereus* ATCC 14579, *Enterococcus faecalis* ATCC 29219, *Escherichia coli* ATCC 25922, *Salmonella typhimurium* ATCC 14028, and *Salmonella enteritidis* ATCC 13076. For each organism, an inoculum was made suspending 24 h pure culture in Mueller Hinton (MH) broth to obtain 10E6 colony forming unit (CFU)/mL according to McFarland scale. The MH agar plates surface was “flood-inoculated” with the bacterial inoculum and prepared for the extract’s evaluation; six-millimeter diameter wells (three for each extract) were aseptically made into the MH agar to contain 60 μL of tested product and 70% ethanol, respectively (as the negative control). Gentamicin was also included as standard antibiotic. The growth inhibition zones diameters in millimeters were measured after 24 h of incubation at 37 °C. Furthermore, the extracts minimum inhibitory (MIC) and bactericidal (MBC) concentrations were established using a broth microdilution method. Two-fold serial dilutions were made in 100 µL broth for each of the two extracts; 5.0 µL of a 24 h 1 × 10^7^ CFU/mL bacterial inoculum were added in each well and incubated for 24 h at 37 °C. MICs values were read as the lowest concentrations able to inhibit the visible growth of bacteria (no turbidity in the well), when compared to the negative control (broth). From each well, 10.0 µL were cultured on MH agar plates for 24 h at 37 °C. MBCs values were read as the lowest concentrations associated with no visible bacterial growth on the agar plates. All these tests were performed in triplicate.

Based on the ratio MBC/MIC, the MIC index was also calculated for each extract to evaluate whether the extract exhibits bactericidal (MBC/MIC ≤ 4) or bacteriostatic (MBC/MIC > 4) effect against the tested bacterial strains [63].

### 3.7. Isolation of Rat Intestinal Epithelial Cells

Intestinal epithelial cell culture was prepared using the method previously described by Evans et al., with some adaptations [64]. Pregnant Wistar female rats, aged 8 to 10 weeks, were sacrificed on the 14th day, following vaginal plug detection, according to the European Union Directive 2010/63/EU [65]. The pregnant uterus was revealed through a transversal abdominal incision, and transferred to a Petri dish with sterile phosphate buffer solution (PBS). Fetuses were collected and immersed in Tyrode solution. After the incision of the abdominal wall, the small intestine was collected from each fetal donor, and washed five times with PBS. Small intestine samples were cut into smaller pieces, and immersed in enzymatic solution, which included 2 mg/mL collagenase type I solution and 0.1 mg/mL dispase type I in HBSS (Hanks’ Balanced Salt Solution), and incubated for 15 min at 25 °C on a magnetic stirrer. After incubation, cells were resuspended by pipetting the cell suspension for 2 min and examined under the inverted microscope to confirm the separation of different tissue components. At this phase, the suspension was composed from muscle fragments, multicellular epithelial aggregates, single cells, and cellular debris. When more than 70% of epithelium of the villosities/crypts were separated, the enzymatic solution was neutralized by adding an equal volume of fetal bovine serum (FCS).

The obtained mixture was allowed to sediment for 1 min, the supernatant containing muscle cells was aspirated, and sediment was resuspended in isolation media composed of DMEM, 10% FCS, and 1% antibiotic-antimycotic. Samples were centrifuged at 3000 RPM for 3 min at 4 °C. Cellular sediment was resuspended in propagation medium DMEM, supplemented with 10% FCS, 1% non-essential amino acids (NEA), 1% antibiotic-antimycotic, 0.25 IU/mL, and 20 ng/mL epidermal growth factor (EGF). Following the evaluation of cell number and viability (0.4% Trypan blue), cells were cultivated on propagation medium and incubated at 37 °C, 5% CO_2_, and relative humidity of 60–90%.

After 48 h of propagation, nonadherent cells were removed by changing the medium, for the adherent cells the medium was renewed every 2 days. The passage of cells continued until the cell culture reached a 70% confluence. The primary cell culture was passaged 4 times before further use in the toxicity study.

### 3.8. Cytotoxicity Assay

To evaluate the cytotoxic effect of 70% ethanolic extracts of *I. helenium* and *G. asclepiadea* roots, MTT assay (3-(4,5-Dimethyl-2-thiazolyl)-2,5-diphenyl-2H-tetrazolium bromide) was performed. In the assay, 96-well culture plates, with a density of 1 × 10^5^ cells/well, and 200 µL of culture medium were used. Rat intestinal epithelial cells were treated with 0.25, 0.5, 1.0, 2.5, 5.0, 7.5, 10.0, and 15.0 mL of extracts, and resulting concentrations in the wells (C1, C2, C3, C4, C5, C6, C7, C8) were calculated according to the total phenolic content of each plant extract, and expressed as μmol GAE/mL extract, as follows: *G. asclepiadea* (C1= 0.0079, C2 = 0.0158, C3 = 0.0315, C4 = 0.0788, C5 = 0.1575, C6 = 0.2363, C7 = 0.3151, and C8 = 0.4726 μmol GAE/mL), and *I. helenium* (C1 = 0.0135, C2 = 0.0269, C3 = 0.0538, C4 = 0.1345, C5 = 0.2689, C6 = 0.4034, C7 = 0.5379, C8 = 0.8068 μmol GAE/mL). Each concentration was tested in triplicate. The negative control was represented by untreated cells.

After 24 h of exposure to the extract, the culture medium was removed from the wells and 100 μL of MTT solution (0.5 mg MTT/mL HBSS buffer) were added to each well. The culture plates were incubated at 37 °C for 4 h. Subsequently, the MTT reagent was removed and 100 μL of dimethylsulfoxide (DMSO, Sigma) was distributed to each well to solubilize the formazan particles. Absorbance of the chromogenic reaction was measured by spectrophotometry with a BioTek Synergy 2 microplate reader (Winooski, VT, USA), at a wavelength of 450 nm. Results were presented as average % of cell viability (2):(%) Viability = (mean sample OD/mean control OD) × 100%,(2)
where OD stands for the optical density value. Cell viability and proliferative capacity of treated cells were compared with the negative control. In addition, for each tested plant extract, the half-maximal inhibitory concentration (IC_50_) value was calculated from the dose response curve obtained using non-linear regression [60].

### 3.9. Statistical Analysis

The obtained data were statistically analyzed using ANOVA GraphPad Prism software, version 6.0 (San Diego, CA, USA). The samples were analyzed in triplicate and quantitative determinations were given as mean ± standard deviation (SD). The compounds under quantification (<LOQ) limits could be only identified. One-way analysis of variance (ANOVA) was conducted, followed by Tukey’s post hoc test, to determine statistical significance between the chemical profile components and the antioxidant, antimicrobial, and cytotoxic potential of the two extracts, considering statistically significant *p* < 0.05. In addition, CORREL function was used to calculate Pearson’s correlation coefficients for the analyzed data, namely total phenolic, flavonoid, and phenolic acids content in tested extracts and the antioxidant, antimicrobial, and cytotoxic activity, respectively.

## 4. Conclusions

To our knowledge, the present study represents the first report in scientific literature regarding the lack of toxicity of *G. asclepiadea* and *I. helenium* ethanolic root extracts on rat intestinal epithelial cells.

The study highlighted the fact that these two extracts represent valuable sources of bioactive compounds with therapeutic potential. Particularly, identification and quantification of amarogentin in *G. asclepiadea* harvested from Romania, is of scientific interest, since the population of other gentian species of high medicinal interest is decreasing worldwide (*G. lutea*, *G. punctata*, *G. dinarica*), positioning it as a promising substitute.

Furthermore, a direct correlation between the biological activities of these plant extracts and their total phenolic and flavonoid concentrations was indicated by the statistical analysis. These data support their ethnomedicinal recommendations in digestive disorders, further studies are intended to develop standardized therapeutic products.

## Figures and Tables

**Figure 1 molecules-27-03560-f001:**
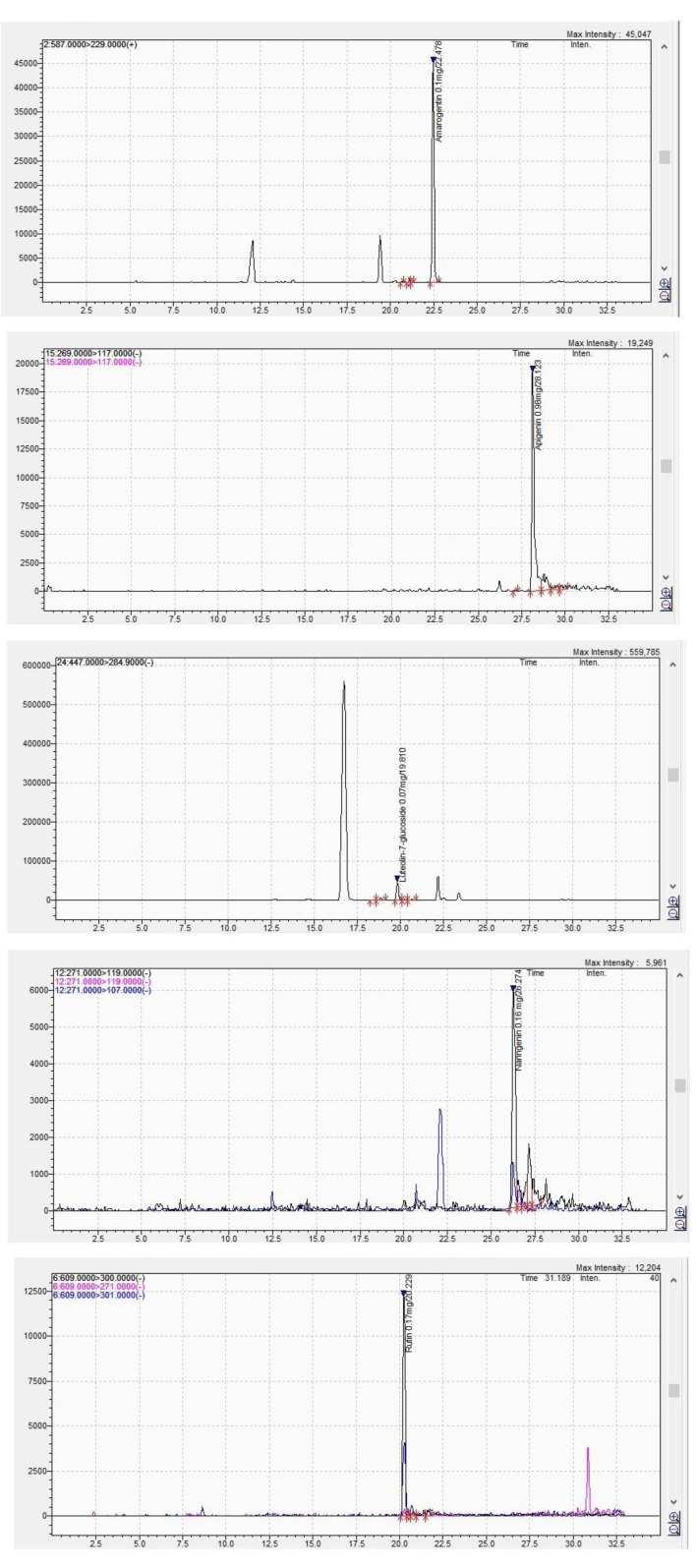
LC-MS chromatogram peaks of *G. asclepiadea* ethanolic extract—amarogentin, apigenin, luteolin-7-*O*-glucoside, naringenin, rutoside (**top** to **bottom**).

**Figure 2 molecules-27-03560-f002:**
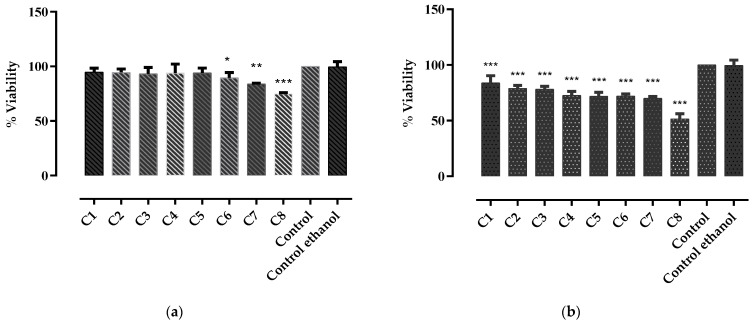
Inhibitory effects of (**a**) *G. asclepiadea* and (**b**) *I. helenium* root extracts on rat intestinal epithelial cells, at eight different concentrations C1–C8 calculated according to the TPC (μmol GAE/mL extract): ranging between 0.0079 and 0.4786 for *G. asclepiadea* extract, and from 0.0135 to 0.8068 for *I. helenium* extract. Control—untreated cells, Control ethanol—cells treated with ethanol. Values are represented as mean ± SD. Statistically significant differences between treated and untreated cells (control): * *p* < 0.05; ** *p* < 0.001; *** *p* < 0.0001.

**Table 1 molecules-27-03560-t001:** Total polyphenolic (TPC), flavonoid (TFC), and phenolic acids (TPA) content of *Gentiana asclepiadea* and *Inula helenium* extracts.

Sample	TPC (g GAE/100 g Dry Plant Material)	TFC (g RE/100 g Dry Plant Material)	TPA (g CAE/100 g Dry Plant Material)
*G. asclepiadea*	2.144 * ± 0.088	0.280 * ± 0.014	0.224 * ± 0.030
*I. helenium*	3.066 * ± 0.041	0.602 * ± 0.016	1.182 * ± 0.017

Note: Each value represents the mean ± standard deviations of three independent measurements. GAE: Gallic acid equivalents; RE: rutin equivalents, CAE: caffeic acid equivalents. * *p* < 0.05 *G. asclepiadea* vs. *I. helenium.*

**Table 2 molecules-27-03560-t002:** The identified and quantified components in the *G. asclepiadea* and *I. helenium* root ethanolic extracts (µg/g dry vegetal material) by the LC-MS analysis.

Compound	Retention Time, min		*m*/*z* and Main Transition		Concentration
	Standard	Separated Compound	Standard	Separated Compound	Value
*G. asclepiadea*					
Caffeic acid	13.8	13.7	179.0 > 135.0	179.0 > 135.0	169 ± 1.2
*trans*-*p*-coumaric acid	17.5	17.6	163.0 > 119.0	163.0 > 119.0	192.8 ± 1.0
Salicylic acid	23.5	23.5	137.0 > 93.0	137.0 > 93.0	<LOQ
Chlorogenic acid	12.0	12.0	353.0 > 191.0	353.0 > 191.0	33.4 ± 0.5
Amarogentin	22.5	22.5	587.0 > 229.0	587.0 > 229.0	27.8 ± 0.3
Apigenin	28.2	28.1	269.0 > 117.0	269.0 > 117.0	18.0 ± 0.7
Chrysin	29.7	29.7	253.0 > 143.0	253.0 > 143.0	<LOQ
Luteolin	26.9	26.8	287.0 > 153.0	287.0 > 153.0	9.6 ± 0.2
Luteolin-7-*O*-glucoside	19.9	19.8	447.0 > 284.9	447.0 > 284.9	22.6 ± 0.5
Quercetin	25.7	25.5	300.9 > 151.0	300.9 > 151.0	<LOQ
Rutoside	20.3	20.2	609.0 > 300.0	609.0 > 300.0	30.8 ± 0.6
Naringenin	26.3	26.3	271.0 > 119.0	271.0 > 119.0	8.0 ± 0.2
*I. helenium*					
Caffeic acid	13.8	14.0	179.0 > 135.0	179.0 > 135.0	234.0 ± 2.1
Chlorogenic acid	12.0	12.2	353.0 > 191.0	353.0 > 191.0	2284.1 ± 11
Chrysin	29.7	29.9	253.0 > 143.0	253.0 > 143.0	<LOQ
Luteolin	26.9	27.5	287.0 > 153.0	287.0 > 153.0	<LOQ
Luteolin-7-*O*-glucoside	19.9	20.5	447.0 > 284.9	447.0 > 284.9	<LOQ
Naringenin	26.3	26.3	271.0 > 119.0	271.0 > 119.0	3.2 ± 0.03
Hesperetin	27.1	27.5	301.0 > 164.0	301.0 > 164.0	<LOQ

Note: <LOQ—below the limit of quantification.

**Table 3 molecules-27-03560-t003:** Antioxidant capacity of *G. asclepiadea* and *I. helenium* root ethanolic extracts.

Sample	FRAP (μmol TE/100 mL Extract)	DPPH IC_50_ (μg/mL)
*G. asclepiadea*	145.23 * ± 3.60	363.7 * ± 0.89
*I. helenium*	629.04 * ± 2.07	173.2 * ± 3.40

Note: Values represent the mean ± standard deviations of three independent measurements, * *p* < 0.05 *G. asclepiadea* vs. *I. helenium.*

**Table 4 molecules-27-03560-t004:** Antibacterial activity of *G. asclepiadea* and *I. helenium* root ethanolic extracts (agar well-diffusion assay).

Zone of Inhibition (mm)						
Sample	*Staphylococcus aureus*	*Bacillus cereus*	*Enterococcus faecalis*	*Salmonella enteritidis*	*Salmonella typhimurium*	*Escherichia coli*
*G. asclepiadea*	15.33 ± 0.47	17.33 ± 0.47	11.33 ± 0.47	11.33 ± 0.47	10.00 ± 0.00	10.33 ± 0.47
*I. helenium*	16.33 ± 0.47	18.00 ± 0.00	8.67 ± 0.47	12.00 ± 0.82	12.00 ± 0.82	10.67 ± 0.47
*G. asclepiadea + I. helenium*	18.33 ± 0.47 ^a,b^	21.00 ± 0.00 ^a,b^	10.33 ± 0.94	12.67 ± 0.47	12.67 ± 0.47	11.33 ± 0.47
Gentamicin	18 ± 0.00 ^a,b^	21 ± 0.00 ^a,b^	17 ± 0.00 ^a,b,c^	18 ± 0.00 ^a,b,c^	17 ± 0.00 ^a,b,c^	17 ± 0.00 ^a,b,c^

Note: Values represent the mean ± standard deviations of three independent measurements. ^a–c^ Means with different subscript letters within a row are significantly different at *p* < 0.05.

**Table 5 molecules-27-03560-t005:** Antibacterial activity of *G. asclepiadea* and *I. helenium* root ethanolic extracts (broth microdilution assay).

MIC Index MBC (μmol GAE/mL)/MIC (μmol GAE/mL)						
Sample	*Staphylococcus aureus*	*Bacillus cereus*	*Enterococcus faecalis*	*Salmonella enteritidis*	*Salmonella typhimurium*	*Escherichia coli*
*G. asclepiadea*	1	1	1	1	1	1
	0.063 × 10^−4^/	0.0315 × 10^−4^/	0.063 × 10^−4^/	0.063 × 10^−4^/	0.063 × 10^−4^/	0.063 × 10^−4^/
	0.063 × 10^−4^	0.0315 × 10^−4^	0.063 × 10^−4^	0.063 × 10^−4^	0.063 × 10^−4^	0.063 × 10^−4^
*I. helenium*	1	0.5	1	0.5	0.5	1
	0.045 × 10^−4^/	0.0225 × 10^−4^/	0.0901 × 10^−4^/	0.045 × 10^−4^/	0.045 × 10^−4^/	0.0901 × 10^−4^/
	0.045 × 10^−4^	0.045 × 10^−4^	0.0901 × 10^−4^	0.0901 × 10^−4^	0.0901 × 10^−4^	0.0901 × 10^−4^

Note: Values represent the mean ± standard deviations of three independent measurements.

**Table 6 molecules-27-03560-t006:** LC-MS mobile phase gradient.

Time, Min	Methanol	Water	2% Formic Acid in Water
0.00	5	90	5
3.00	15	70	15
6.00	15	70	15
9.00	21	58	21
13.00	21	58	21
18.00	30	41	29
22.00	30	41	29
26.00	50	0	50
29.00	50	0	50
29.01	5	90	5
35.00	5	90	5

**Table 7 molecules-27-03560-t007:** LC-MS standards identification parameters.

Name of Standard	Retention Time, min	*m*/*z **, and Main Transition	MRM	Other Transitions
Caffeic acid	13.8	179.0 > 135.0	Negative	179.0 > 134.0 179.0 > 89.0
*trans*-*p*-coumaric acid	17.5	163.0 > 119.0	Negative	163.0 > 93.0
Salicylic acid	23.5	137.0 > 93.0	Negative	137.0 > 75.0 137.0 > 65.0
Chlorogenic acid	12.0	353.0 > 191.0	Negative	
Amarogentin	22.5	587.0 > 229.0	Positive	
Apigenin	28.2	269.0 > 117.0	Negative	
Chrysin	29.7	253.0 > 143.0	Negative	253.0 > 119.0 253.0 > 107.0
Luteolin	26.9	287.0 > 153.0	Positive	
Luteolin-*7*-*O*-glucosid	19.9	447.0 > 284.9	Negative	
Quercetin	25.7	300.9 > 151.0	Negative	300.9 > 121.0
Rutoside	20.3	609.0 > 300.0	Negative	609.0 > 301.0 609.0 > 271.0
Naringenin	26.3	271.0 > 119.0	Negative	271.0 > 107.0
Hesperetin	27.1	301.0 > 164.0	Negative	301.0 > 136.0 301.0 > 108.0

Note: * *m*/*z* = mass-to-charge ratio.

**Table 8 molecules-27-03560-t008:** LC-MS standards quantification parameters.

Name of Standard	Concentration Range, mg/mL	Calibration Curve Equations	Correlation Factors	LOD, μg/mL	LOQ, μg/mL
Caffeic acid	0.11–1.10	A = 4 × 10^7^ × c − 319,689	0.9998	3.20	4.80
*trans*-*p*-coumaric acid	0.16–1.60	A = 3 × 10^7^ × c + 291,065	0.9993	1.90	3.90
Salicylic acid	0.16–1.60	A = 4 × 10^7^ × c + 44,120	0.9997	1.50	2.00
Chlorogenic acid	0.13–1.30	A = 2 × 10^8^ × c − 269,699	0.9997	5.00	8.00
Amarogentin	0.10–1.00	A = 3 × 10^8^ × c − 36,887	0.9997	5.00	7.00
Apigenin	0.10–0.98	A = 2 × 10^8^ × c + 15,916	0.9999	0.20	0.30
Chrysin	0.10–1.00	A = 1 × 10^8^ × c − 82,818	0.9997	3.00	5.00
Luteolin	0.01–0.10	A = 2 × 10^8^ × c − 2295.4	0.9977	0.05	0.07
Luteolin-*7*-*O*-glucosid	0.07–0.70	A = 1 × 10^9^ × c − 700,317	0.9990	3.00	4.00
Quercetin	0.09–0.91	A = 5 × 10^7^ × c − 9556	0.9964	0.80	1.10
Rutoside	0.17–1.70	A = 2 × 10^8^ × c − 191,937	0.9996	4.00	6.00
Naringenin	0.16–1.60	A = 3 × 10^8^ × c − 43,443	0.9999	0.60	0.90
Hesperetin	0.10–1.00	A = 6 × 10^7^ × c − 49,247	0.9974	3.00	5.00

Note: A = Area; c = concentration (mg/mL).

## Data Availability

Not applicable.

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
