# Peer review of "Biological Activities and Chemical Profile of Gentiana asclepiadea and Inula helenium Ethanolic Extracts"

_molecules, 2022, doi:10.3390/molecules27113560_

Round 1
Reviewer 1 Report
The paper “Biological Activities and Chemical Profile of Gentiana asclepi- adea and Inula helenium Ethanolic Extracts” is focused to the investigation of the antioxidant, antimicrobial, and cytotoxic potential of ethanolic extracts obtained from Gentiana asclepiadea L. and Inula helenium L. roots, in relation to their chemical composition.
The paper is interesting, the abstract and the introduction are well explained but a good check of English should be done. As well as for the whole manuscript.
The results obtained using spectrophotometrical methods for the quantification of 124 TPC, TFC and TPA content need to be more discussed and compared with other bibliography as well as for the results of the Antioxidant activity.
The results of the in vitro antimicrobial activity are well organized but a greater comparison with previous studies is needed.
Reviewer 2 Report
The comments are as follows.
- The particle size and moisture content of dried material should be provided.
- Why the authors chose maceration as extraction technique? How are the extraction conditions determined? Is not the 14-days extraction too long, since nowadays there are more efficient extraction techniques for shorter extraction time?
- The authors identify only 12 and 7 compounds in Gentiana asclepiadea and Inula helenium ethanolic extracts, respectively. Is it possible to tentatively identify more compounds since HPLC-MS is used?
- Lines 215-219. This sentence is too long. Also, IC50 values of 177.10 and 364.73 are not same as in the Table 4.
- In my opinion, a discussion about the relation between the compounds identified in the extract and the different bioactivities exerted should be included in order to be considered for publication. Additionally, for a better understanding of the relation between the compounds in the extracts and the bioactivities exerted, comprehensive chemical analysis should be provided.
- Are really necessary the citations of 9 articles of the coauthor dr. Daniela Hanganu?
- Are really necessary the citations of 110 references for a research article? Please, reduce them considerably.
- Please, revise the text to correct some misprints.
Round 2
Reviewer 2 Report
1. The authors are encouraged to reduce the number of references considerably.
Author Response
Dear Editor and Reviewer,
On behalf of our co-authors, I would like to submit the revised form of the manuscript (Molecules-1742660) entitled “Biological Activities and Chemical Profile of Gentiana asclepiadea and Inula helenium Ethanolic Extracts”, sent for publication in the “Molecules” journal, together with the response to the reviewer’s comments after the second round of review, which are found below. The suggested changes were performed with “Track changes”, for the benefit of editors and reviewers.
Please see the attachment.
